



# No critical slowing down in the Atlantic Overturning Circulation in historical CMIP6 simulations

Maya Ben-Yami[1,2], Lana Blaschke[1,2], Sebastian Bathiany[1,2], and Niklas Boers[1,2,3]

[1]Earth System Modelling, School of Engineering and Design, Technical University of Munich, Munich, Germany
[2]Potsdam Institute for Climate Impact Research, Potsdam, Germany
[3]Department of Mathematics and Global Systems Institute , University of Exeter, Exeter, UK

**Correspondence:** Maya Ben-Yami (maya.ben-yami@tum.de)

**Abstract.** The Atlantic Meridional Overturning Circulation (AMOC) is a key component of the Earth's climate system, and is
theorized to have multiple stable states. Critical slowing down (CSD) can detect stability changes in Earth system components,
and has been found in sea-surface temperature (SST) based fingerprints of the AMOC. We look for CSD in simulations from
27 models from the sixth Climate Model Intercomparison Project (CMIP6). We calculate three different CSD indicators for
the AMOC streamfunction strengths at 26.5°N and 35°N, as well as for a previously suggested AMOC fingerprint based on
averaging SSTs in the subpolar gyre region. We find mixed results. Most models do not have a statistically significant sign
of CSD, and no model has a conclusive sign of CSD in all ensemble members. However, some models exhibit a number of
significant increases in the CSD indicators of the streamfunction strength that is highly unlikely to occur by chance ($p<0.05$). In
addition, the number of significant increases in the AMOC SST fingerprint are as would be expected by random chance. Since
we do not know whether or not the AMOC in these models is approaching a critical transition, we cannot deny or confirm the
validity of CSD for detecting an upcoming AMOC collapse. However, we can confirm that the AMOC SST fingerprint is not
prone to false positives.
# 1  Introduction
The Atlantic Meridional Overturning Circulation (AMOC) is a vital part of the Earth's climate system. It consists of a system of
currents that transport large amounts of heat and salt northward near the surface of the Atlantic Ocean. The AMOC is predicted
to weaken under anthropogenic global warming (Lee et al., 2021), with substantial effects on the climate (Bellomo et al.,
2021; Liu et al., 2020). There is some observational and paleoclimate evidence that the AMOC has already weakened (Caesar
et al., 2021; Zhu et al., 2023), although the decline is controversial and likely still consistent with natural variability (Kilbourne
et al., 2022; Latif et al., 2022). The AMOC has also been identified as a potential climate tipping element (McKay et al.,
2022; Boers et al., 2022), with support for bistability coming from paleoclimate proxy records (Henry et al., 2016), theoretical
studies (Stommel, 1961; Cessi, 1994), as well as from experiments with some general circulation models (Rahmstorf, 2002;
Hawkins et al., 2011; Liu et al., 2017; Jackson and Wood, 2018; Westen et al., 2024). However, the AMOC appears to be
monostable in other models, and the question of alternative stable AMOC states in comprehensive climate models remains
debated (Jackson et al., 2023a). Roughly, the AMOC's bistability is theorised to arise from a key positive feedback in the



system, the salt advection feedback. A weaker AMOC leads to less northward salt transport and thus reduced surface density
in the regions of north Atlantic deep water (NADW) formation. The reduction in NADW formation in turn leads to an even
weaker AMOC, continuing until an alternative weak AMOC state is reached. A question of intense debate is whether or not
the AMOC could undergo such an abrupt collapse under future anthropogenic climate change (Mckay et al., 2022; Boers
et al., 2022). The 6th assessment report (AR6) of the International Panel on Climate Change (IPCC) concludes that there is
medium confidence that such a collapse will not happen before 2100 (Arias et al., 2021). This conclusion is in large part based
on results from complex climate models, and in particular on the experiments of the sixth Climate Model Intercomparison
Projects (CMIP6). Yet the complexity and extent of the AMOC makes it challenging to simulate, and complex models can fail
to capture key processes which could lead to a warming-induced collapse (Jackson et al., 2023b; Liu et al., 2017). When the
small number of experiments that modelled a global-warming induced AMOC collapses are used to estimate the temperatures
at which a collapse might happen, these range from 1.4 to 8 °C, in practice encompassing the full scope of future warming
(Mckay et al., 2022).
Another approach to understanding the stability of the AMOC is to use its characteristic statistical properties from the
perspective of nonlinear dynamical systems. In particular, when a system approaches a bifurcation-induced transition to a
different stable state it undergoes a process called Critical Slowing Down (CSD), in which the rate of the system's return
to equilibrium after perturbations weakens and eventually approaches zero. Statistical indicators of CSD, such as increasing
variance or autocorrelation, are a measure of the stability of the system. In extreme cases they can signal an approaching
transition, and have thus sometimes been called early-warning signals (Scheffer et al., 2009). Boers (2021) (hereafter B21)
studied sea surface temperature (SST)- and salinity-based observational proxies of the historical AMOC strength, and found
a significant signal of CSD in these proxies. B21 also applied this analysis to time series of AMOC streamfunction strengths
and SSTs from the historical runs of the fifth Climate Model Intercomparison Project (CMIP5), but found no consistent sign
of CSD in those runs. In this study we repeat this CSD analysis on a range of more recent state-of-the-art models from CMIP6,
expand it to other CSD indicators and AMOC fingerprints, and look for patterns in the results.
There are two main reasons why such an analysis is of interest. The first is that to trust the projections of climate models we
have to understand their accuracy and biases, and one of the best ways to do that is to compare their results to observations of
the climate. Signs of CSD are consistently present in observational proxies that have been suggested to represent the AMOC,
and so we can evaluate models by the presence or lack of CSD in the corresponding model proxy time series from the historical
runs.
The second reason is to validate the usefulness of these SST- and salinity-based proxies for detection of CSD for the AMOC.
In an ideal scenario, one would calculate CSD indicators directly from the time series that characterises the system: the AMOC
streamfunction strength. However, CSD detection requires long timeseries, and direct observations of AMOC strength only start
in 2004 (Frajka-Williams et al., 2021). Studies have thus identified multiple AMOC fingerprints that use long-term observation-
based time series that in models have been shown to be correlated – at different strength and different lags – with the AMOC
strength (Jackson and Wood, 2020).





One of the most commonly used AMOC fingerprints is based on the SSTs in the subpolar gyre (SPG), and in this work
will be called the AMOC SST index (Caesar et al., 2018; Rahmstorf et al., 2015). Although there are some concerns about
this fingerprint (Little et al., 2020; Zhu et al., 2023), studies have found correlations between this index and the AMOC
streamfunction strength that are sufficient for understanding the historical AMOC trend (Caesar et al., 2018; Jackson and
Wood, 2020). Thus the observed decrease in the SST index likely indicates a decrease in the strength of the AMOC (Caesar
et al., 2018). However, so far no study has examined if this or other fingerprints can be used to detect CSD in the AMOC
under anthropogenic forcing. The best evidence for using the SST index to detect CSD would be to consistently see CSD in
the fingerprint in models where the AMOC then subsequently undergoes a warming-induced collapse. Unfortunately, there are
very few models in which the AMOC collapses due to global warming, and there are no studies that test the AMOC SST index
in this manner.
In this work, we calculate CSD indicators for a large range of CMIP6 models. We examine our results to see if they can be
used to evaluate the climate models, and whether they validate the use of the AMOC SST Index for CSD detection.

## 2    Results

### 2.1    AMOC timeseries slopes

In this work, the AMOC streamfunction for the different CMIP6 models is computed in the same way as in Menary et al. (2020).
To get the streamfunction for a larger number of models, the oceanic meridional velocities are used to calculate the overturning
stream function directly for each model. For the streamfunction strength we take the maximum strength of overturning at
latitudes 35°N and 26.5°N. These time series will in the following sometimes be abbreviated as S35 and S26, respectively. The
AMOC SST Index is calculated following Caesar et al. (2018) but with the SPG area simplified as in Menary et al. (2020). The
index is the average SSTs in the area between 41° and 60° N and 20° and 55° E, minus the average global SSTs. The AMOC
SST Index will sometimes be shortened to ASSTI. The list of CMIP6 models and their ensemble members can be found in
Table C1.
As expected, the slope of the AMOC streamfunction strength at the two different latitudes is highly correlated (Fig. 1).
Throughout this work, slope stands for the slope of a fitted linear trend. The correlation of the streamfunction slope with the
AMOC SST Index slope is present in CMIP6 (Fig. 2), as seen in other studies in CMIP5 (Caesar et al., 2018; Menary et al.,
2020), and has an R value of 0.7 and 0.67 for 26.5°N and 35°N, respectively. As discussed in many previous works, a large part
of the AMOC time series in CMIP6 have a positive slope, i.e. an increasing AMOC strength in the historical period (Menary
et al., 2020; Weijer et al., 2020). For the SST Index this is in opposition to the slope in the observed SSTs, which is around -0.5
C°/Century. This increase in AMOC strength in CMIP6 is mostly present in the time span of 1850-1985, and has in part been
attributed to too-strong aerosol forcing (see Menary et al. (2020); Robson et al. (2022); Weijer et al. (2020) for discussion of
this forcing and for full AMOC time series). In addition, other work has shown that the large scatter in slopes is also caused
by internal AMOC variability (Bonnet et al., 2021). Indeed, although for each model the ensemble members have the same
forcing, in some cases there is a large spread in ensemble member slopes (Fig. C1).





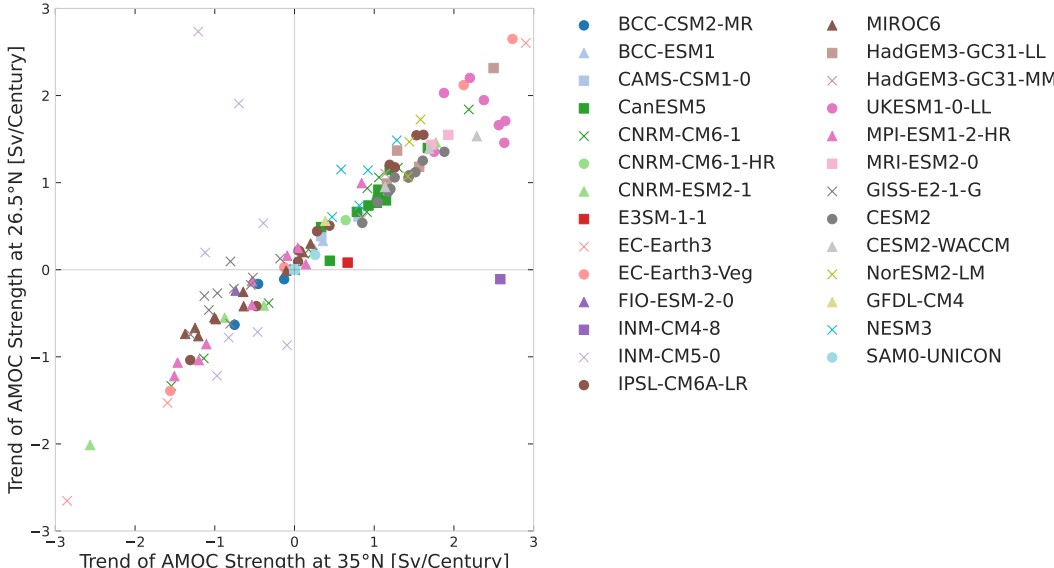

**Figure 1.** slope of AMOC streamfunction strength at 35°N vs 26.5°N for the individual ensemble members of 27 CMIP6 models. For each model all ensemble member slopes are shown using the same coloured symbol.

## 2.2 AMOC CSD indicators

Following B21 the main CSD indicator we will use is the restoring rate $\lambda$. Whilst it is common to use the variance and autocorrelation of a timeseries as CSD indicators, they can result in a false positives or negatives when the statistics of the external conditions change. The restoring rate $\lambda$ as introduced by B21 is estimated under the assumption of non-stationary correlated noise driving the system, and thus avoids biases induced by non-stationary noise (see Methods for details). When a system is in a stable equilibrium state, $\lambda$ is negative, and it increases toward 0 from below as a critical transition is approached. In addition to $\lambda$, we also calculate the variance and autocorrelation of the time series.

Since we are dealing with a large number of time series, statistically $\lambda$ is likely to increase in some cases even without any underlying CSD; this is the so-called multiple comparisons issue. We approach this problem from two different angles. The first is that in the case of true AMOC CSD, it is likely that both streamfunction strengths at the two latitude bands, and possibly the AMOC SST Index, will have an increase in $\lambda$, as well as in the other two classical CSD indicators (variance and autocorrelation). This will be discussed later in this section. The second is statistical significance, which is tested individually for each time series: in particular, for an AMOC time series of given variance and autocorrelation function, how likely is it to have the given increase in $\lambda$? We generate 1000 Fourier surrogates for each AMOC time series, and use those to calculate a p-value for the slope of $\lambda$ in each time series. Whilst we define p≤0.05 as a significant signal, we also use the p-values themselves to compare the models. Note that we calculate the surrogates from the original AMOC time series and not the CSD indicator time series, which is a more conservative approach than followed in B21 (see Ben-Yami et al. (2023) for details).



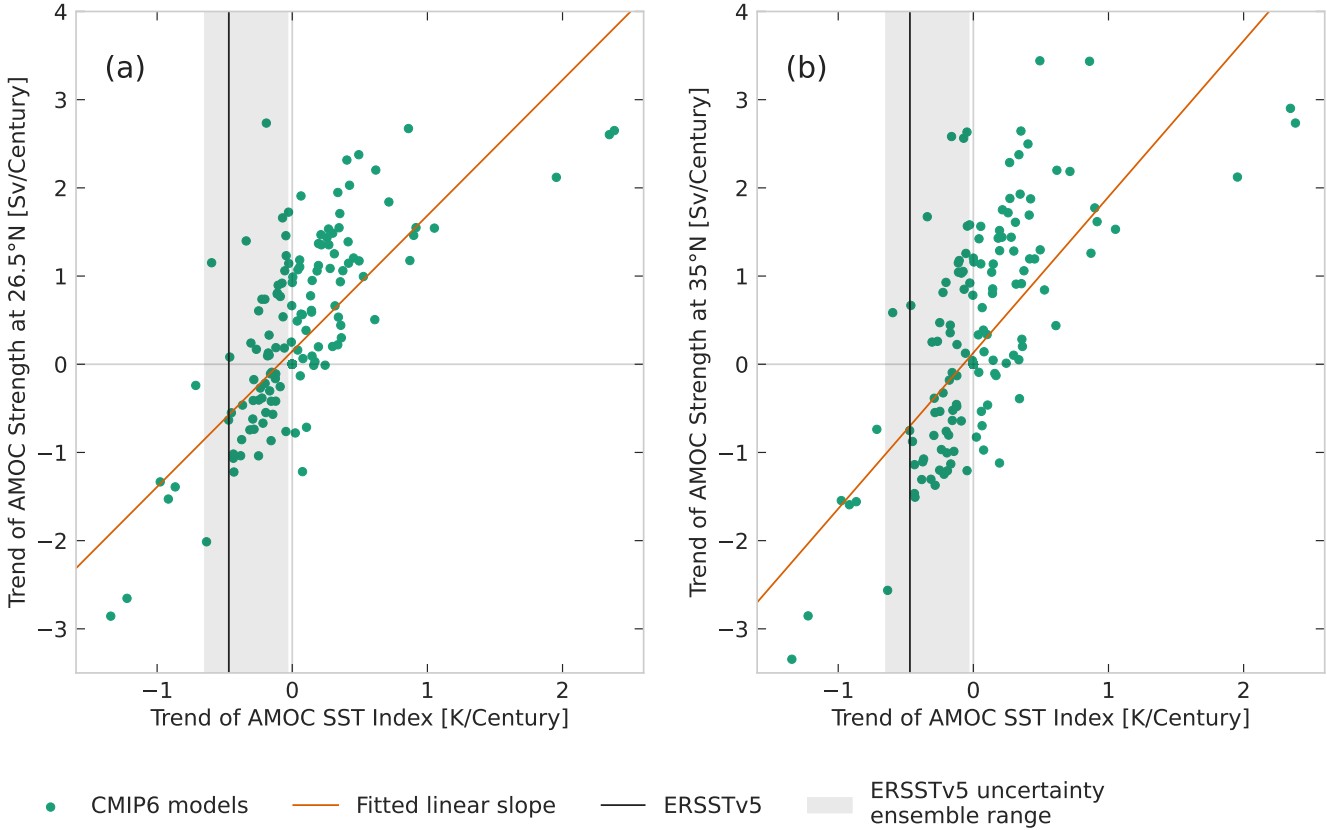

**Figure 2.** slope of AMOC streamfunction strength at 26.5°N (a) and 35°N (b) vs. the slope of the AMOC SST Index. All individual ensemble member slopes of 27 CMIP6 models are shown using the same turqoise dot to emphasize the correlation. The slopes of the linear regressions (orange lines) are 1.54 and 1.77 $SvK^{-1}$ for 26.5°N and 35°N, respectively. The black line is the slope in the ERSSTv5-derived observational AMOC SST Index, with the uncertainty range from the dataset's 1000 member uncertainty ensemble.

Out of the 399 time series for both streamfunction strengths and SST Index for the 27 models, 36 have an increasing slope
in the restoring rate $\lambda$ with p-value below the 0.05 significance level (Fig. 4). However, only 4 ensemble members have a
significant $\lambda$ increase in both streamfunction strengths (CanESM5 r7, MIROC6 r8 and CESM2 r3 and r8, see Fig. 4). There
is only one case in which both the SST Index and a streamfuntion strength have coinciding significant positive $\lambda$ slopes
(CanESM5 r10).
In addition, if the significant $\lambda$ increases were due to AMOC CSD, we might expect to see them consistently across all
ensemble members in a model. The result here is again ambiguous: the significant increases are definitely concentrated amongst
some models, with 11 models (96 ensemble members) having none at all. For example, at one end, IPSL-CM6A-LR has 10
ensemble members (30 time series) but no significant $\lambda$ increase, whilst at the other end, CanESM5 has 10 ensemble members
and 8 significant $\lambda$ increases (about a fourth of its 30 time series). The fraction of significant $\lambda$ increases for each model are





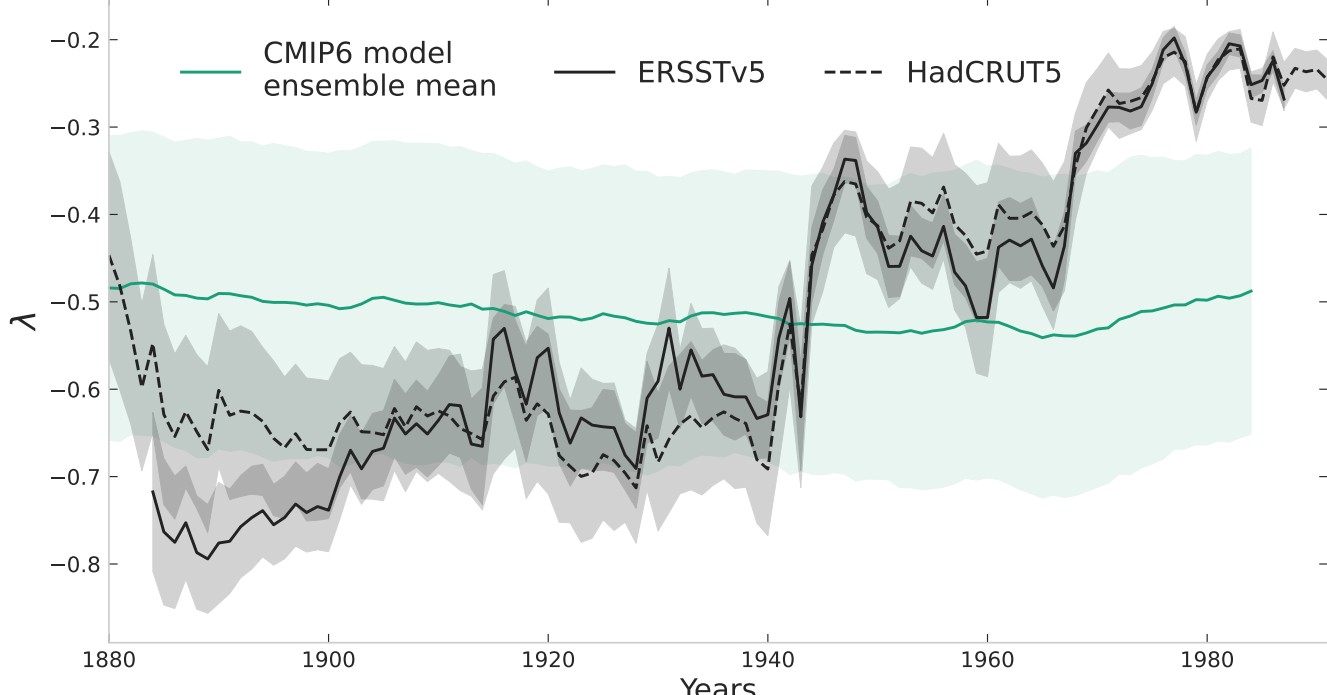

**Figure 3.** Recovery rate $\lambda$ in the observed and modelled SST Index. The mean of the $\lambda$ time series from CMIP6 models (turqoise line) is calculated based on the "1 ensemble member, 1 vote" principle. The one standard deviation of the ensemble is also shown (light turqoise shading). The $\lambda$ timeseries of the SST Index calculated from the SST observational datasets ERSSTv5 (solid black) and HadCRUT5 (dashed black) are also shown with their respecitve uncertainty standard deviations from Ben-Yami et al. (2023).

shown in Figure C2 in the SI. Similarly, when we look beyond the 0.05 hard limit and look at the value of p, we can see that
some models have much smaller p-values than others (Fig. C3).
The next step in testing the validity of a CSD signal is to check the agreement between different CSD indicators. We also
calculate the variance and autocorrelation of the time series, and check for their significant increases following the same
methods. Out of the 399 time series, there are 73 time series in total with any sort of significant CSD indication (Fig. 5). Of
these, 47 have only one significant indicator, 20 have two, and 6 have all three (Table B1). These last six all occur in different
models and all except one occur in the streamfunction at 35°N. Again, there is an uneven distribution of these indicators among
the models (see Fig. C4 for the fraction of significance per model).
Because our definition of statistical significance is a p-value below 0.05, even if all time series are completely random
we would still expect about 5% that fit our criteria for statistical significance. Thus, we need to take that into account and
calculate probabilities if we want to know whether our results are random or not (see Methods for details). The different CSD
indicators and the AMOC strengths and index are interdependent (Figs. C5, 6d, C6d and C7d), and thus we consider each
indicator-strength combination separately when calculating probabilities.



When the results for all models are considered together, the number of significant increases in all three CSD indicators is
significant for the two streamfunction strengths. However, the results for the AMOC SST Index are not significant, and in fact
the probability is only 10-15% because the number of significant ASSTI time series is less than the expected number, not more.
When we consider the probability for each model individually, there are only five model indicator-strength combinations that
are below the 0.05 probability level: CanESM5 $\lambda$ S35, CESM2 $\lambda$ S35, CESM2 AR1 S26, HadGEM3-GC31-LL AR1 S35 and
MIROC6 AR1 S35 (Table B2). Note however that these probabilities are highly conservative (see Methods).
However, even the models with the statistically unlikely result still have ensemble members with an inconclusive or nonex-
istent signal, so it is uncertain if the fact that the result is unlikely to happen by chance is a definite sign of CSD. If the results
were more consistent within models, we could try to formulate necessary criteria for CSD. But to match the observations, we
would need to see a significant increase in all three of the CSD indicators in the AMOC SST Index, something which is not
present in any of the models' ensemble members. If we can interpret the behavior of the AMOC in historical CMIP6 simu-
lations as a conservative null model (since no model has an AMOC collapse before 2100), our results suggest that the CSD
detected in observation-based AMOC fingerprints (Boers, 2021; Ben-Yami et al., 2023) would be highly unlikely to occur by
chance.
Whilst the list of time series with a significant $\lambda$ slope changes slightly when the CSD analysis parameters are changed (see
Figures C8, C9, C10, C11 and Tables C2 and C3), the statistics and inconclusiveness of the results remain the same.

## 2.3    Correlation of CSD indication with other parameters

Finally, we will try to make some sense of the chaotic list of ensemble members and indicators by looking for correlations
between the p-values of the different CSD indicator slopes and other properties and observables. This could, for example,
explain why only a few ensemble members in each model show CSD, since we have seen the AMOC slope also varies between
ensemble members. We compare the p-values with other p-values, with the slope of the historical AMOC, and with the mean
AMOC strength.
Unfortunately, there is almost no correlation between p-values of $\lambda$-slopes and these properties (Fig. 6). The same is also
true for the AR1 and variance (Figs. C6 and C7). The only exception is the correlation between the p-values of the two
streamfunction strengths, which is strongest for the variance but also present for the AR1 and $\lambda$ slopes (Figs. 6d, C6d and
C7d). This correlation is also present for the ensemble means (Fig. C12). This is not very surprising, as the streamfunction
strengths at different latitudes are correlated to some extent. For the ensemble means we also find a negative correlation between
each streamfunction strength p-value and the corresponding mean strength (Fig. C12g-i). However, this correlation disappears
when two outliers with very low mean streamfunction strength are removed.

## 3    Discussion and Conclusions

We have analysed CSD indicators in time series representing AMOC strength at different latitudes as well as an SST-based
AMOC fingerprint in the historical runs of CMIP6. We have chosen to focus on the historical time period so that we can




**Figure 4.** Overview of ensemble members and the occurrence of significant increases in $\lambda$ in AMOC strength and the SST fingerprint. Each model is represented by a circle which is cut into a number of slices corresponding to the ensemble members analysed in this study. These slices are also cut into three layers, each representing one of either the AMOC streamfunction strength time series or the AMOC SST Index. So a model with 10 ensemble members will have 10x3=30 slice layers in the circle corresponding to AMOC time series. For each time series, if its $\lambda$ is significantly increasing at 0.05 confidence level, the corresponding slice layer is coloured, otherwise it is left grey. The order from outside in is: S26.5N (turqoise), S35N (purple) and AMOC SST index (pink). The name of the model and the number of ensemble members are printed above each circle. The first ensemble member (r1) is always the one above the +90° line from the top (3 o'clock), and the ensemble members then proceed counterclockwise. For example, for CanESM5 the SST Index (pink) has a significant slope in $\lambda$ in member r10, and the S26.5 index in members r7, r8 and r10. The corresponding list can be found in Table C4.





**Figure 5.** Number of indicators with a significantly increasing slope in the ensemble members' AMOC strengths and SST fingerprints. The structure of the figure is the same as Figure 4, but instead of representing a significant increase in $\lambda$, the colours now represent different numbers of indicators with a significant increase. These are all three (dark purple), two out of three (pink) and just one (light pink).

compare our results to the observed AMOC SST Index, where there is a significant increase in the three main CSD indicators (Ben-Yami et al., 2023; Boers, 2021). With regards to this comparison, the results are conclusive: no model agrees with the observed indication of CSD in the SST Index (Figs. 3, 5). There are only four significant increases in $\lambda$ in the SST Index (out of 133 time series), and similarly the number of significant increases in the AR1 and variance are also not unlikely (Table B3).

The results are less conclusive for CSD in the streamfunction strengths. We should first note that none of the models used in this study have an abrupt collapse of the AMOC before 2100 in any of the future warming scenarios (Weijer et al., 2020).







**Figure 6.** Correlations between the p-values of $\lambda$ slopes and other quantities. S26.5, S35 and ASSTI stand for the AMOC streamfunction strengths at 26.5°N and 35°N and the AMOC SST Index, respectively. The x-axis is always the p-values of the slope of $\lambda$ calculated from S26.5 (a,d,g), S35 (b,e,h) or ASSTI (c,f,i). This is plotted against the slope of the AMOC time series itself (a-c), the p-values of the slope of $\lambda$ calculated from another of the time series (d-f) and the mean strength of the time series (g-i). All panels show the individual ensemble members of 27 CMIP6 models. For each model all ensemble member slopes are shown using the same coloured symbol. Where appropriate darker grey lines mark the 0.05 significance region.

Thus, even if the AMOC does collapse after 2100 (see Romanou et al.), we may still not be able to detect CSD 200 or more
years in advance. We therefore would not necessarily expect to see CSD in the streamfunction strength time series, even if the
AMOC in a given model is bistable and ultimately can undergo a warming-induced abrupt transition.
However, in some models we do see signs of CSD in the historical period, even if the results are not consistent enough to
manifest a conclusive signal. Using a conservative probability calculation, we find four models that have a number of significant
increases that are statistically significant: CanESM5, HadGEM3-GC31-LL, CESM2 and MIROC6 (Table B2). The latter two
of these also have one ensemble member with what we would classically expect for an indication of CSD: significant increases





of all three CSD indicators in the streamfunction strength at 35°N, and an increase in two out of three CSD indicators in the
streamfunction strength at 26.5°N. Indeed, out of the six timeseries with significant increases in all three CSD indicators, five
are of the streamfunction strength at 35°N. If this is due to the AMOC truly destabilizing in these models, it would make sense
that the signal would be stronger in the more northern latitude, as the destabilization usually originates in the north Atlantic.
The situation is complex, but we can narrow our discussion to four major questions:
1. What is happening in the models with no indication of CSD?
2. What is happening in the models with a significant indication of CSD, but only in a fraction of the ensemble members
and time series?
3. Why is there no CSD in the modelled SST index, as opposed to the observed SST index?
4. What do our answers to questions (1-3) imply for the utility of the SST Index for detection of CSD?
Question (1) is the easiest to answer. We have identified four models where the signs of CSD are significant, but our
probability calculation is conservative, and for some models it is difficult to say if the indication of CSD is present or not.
But for around 20 out of the 27 models it seems like there's definitely no indication of CSD. The most straightforward reason
would be that there is simply no CSD occurring in these models, either because the abrupt transition is too far in the future,
or the AMOC is not destabilizing at all in these model runs. This does not mean that the models are correct - the AMOC is
notoriously hard to simulate in complex climate models (Jackson et al., 2023b), and the CMIP6 models may have AMOC
systems that are too stable (Valdes, 2011; Liu et al., 2017). In either case the models simply cannot help us confirm or reject
the validity of observed AMOC CSD.
Answering question (2) is much more difficult, because in those four models the signs of CSD are significant but not
conclusive. There are a few different possible explanations for this:
(a) In these models, the CSD is real only in the ensemble members that show a strong signal in the indicators. For example,
in a small number of ensemble members the phase of the internal variability is such that the AMOC is undergoing CSD,
and that is detected in the streamfunction strength. In the other ensemble members the AMOC is stable. This is unlikely,
since all ensemble members in a given model have the same forcing.
(b) In these models, the CSD is detected only in those ensemble members, but is real in all of the ensemble. As the AMOC
is a complex high-dimensional system, it is possible that the one-dimensional signals considered here cannot properly
capture the changes in its stability. Alternatively, the CSD signs could be masked by other, unrelated processes such as
effects of the atmospheric circulation. This could make it difficult to detect CSD even in the streamfunction strength time
series, and so only a small fraction of the time series would show a signal even if the AMOC is destabilizing in all the
members.
(c) There is no CSD in any of the model ensemble members. Since the signs of CSD are not conclusive, there is also the
possibility that something else is causing those signals in the models. The likelihood of so many signs of CSD in a





completely random system is very low, but it is possible that some underlying processes make it more likely to have
increases in variance, autocorrelation, etc. in those particular models. This would mean that our method of calculating
statistical significance lacks important priors.
There are two ways in which the above answers to question (2) would affect the AMOC SST Index:
– In cases (a) and (b), the indicator slopes in the streamfunction strength are at least a sufficient condition for CSD. Thus
if one of those cases is true, the SST Index would prove to be a bad measure of AMOC CSD, as it doesn't indicate any
CSD when the streamfunction strengths do. It is possible in that case that the SST Index is simply a weaker measure
influenced by too many other factors, not an invalid one - e.g. if the approaching collapse is after 2100, perhaps the signs
of CSD in the SST Index start to emerge later.
– Case (c) is in some ways more favourable for the utility of the SST Index. If the signs of CSD in the streamfunction
strengths are a false alarm, we would at least know that the SST index is not likely to show that particular type of false
positive.
It is not possible to discern which of the above cases is (part of) the correct answer. To do that would require knowing if the
AMOC is becoming less stable in these historical runs. Although there is no abrupt collapse in these models before 2100 in the
SSP scenarios, that is not a guarantee that it would not happen later. While one option is that the AMOC would collapse after
2100, it is also possible that the AMOC becomes unstable under 1850-2014 conditions, but then the specific SSP scenarios
avoid its collapse in the 21st century.
Whichever case is true, there is still one definite result for the AMOC SST Index. The streamfunction strengths are by far
the more direct measure of the AMOC. Thus, if we had seen a large number of significant increases in the SST Index but none
in the streamfunction strengths, we would have had to conclude that the SST Index has many spurious false positives. As it
is, in the CMIP6 models the SST Index follows the statistics we would expect from random time series. Although this does
not confirm that the SST Index would be able to detect AMOC CSD, it does show that it is not overly sensitive to other ocean
processes and could perhaps be quite a conservative fingerprint for measuring CSD. In addition, there is no indication of CSD
in the AMOC SST Index in the runs identified by Swingedouw et al. (2021) as having an abrupt collapse of the SPG in the
21st century (r1i1p1f1 from CESM2-WACCM, MRI-ESM2-0, NORESM2-LM and CESM2). This is an indication that using
the AMOC SST Index to detect CSD in the AMOC would not produce a false warning in case of a collapse of only the SPG.
Finally, it is not only uncertain how well the models represent the AMOC as a whole, but also how well they represent the
connection between its components (Jackson et al., 2023b). In particular, one of the reasons for using the AMOC SST Index
for the detection of CSD relies on a physical connection of those local SSTs to the stability of the AMOC. One of the main
identified restoring forces of SPG perturbations is the negative feedback cycle in which heat loss leads to increased deep water
formation, which in turns leads to a stronger circulation and more heat transport into the SPG (Menary et al., 2015; Sun et al.,
2021). This restoring force is intrinsically connected to the AMOC, and if the AMOC becomes less stable, it takes longer for
the ocean circulation transporting heat into the SPG to restore from perturbations, and thus the SSTs in that region exhibit CSD.
However, it is not certain that these physical mechanisms will be present in CMIP6 models. For example, the extent, location,



and even existence of deep water formation in the SPG differs between models (Jackson et al., 2023b). Conclusive statements
on the validity of the AMOC SST Index would thus require an in-depth analysis of the physical processes, which is beyond the
scope of this work.
Future work on AMOC CSD in complex climate models should focus on scenarios where the AMOC undergoes a warming-
induced abrupt transition. In those cases the utility of different indicators and fingerprints for CSD detection could be analysed
in detail, and would give important context to our results.
In conclusion, we have found that the historical CMIP6 experiments do not agree with the observational records on the
indication of CSD in the AMOC SST Index. There are some models that show significant signs of CSD in the AMOC stream-
function strength, but those signs are not consistent enough to make for a conclusive result. Whilst our results cannot confirm
or deny the utility of the AMOC SST Index for detection of AMOC CSD, they do indicate that the index is not prone to false
positives, which strengthens the significance of the observational AMOC CSD.
*Code availability.*  All code used to analyse the data and generate figures will be uploaded at https://github.com/mayaby.
*Data availability.*  All CMIP6 data used in this study is available online on the ESGF database (https://aims2.llnl.gov). The HadCRUT5
dataset is available at https://www.metoffice.gov.uk/hadobs/. The ERSSTv5 operational dataset is available at
https://psl.noaa.gov/data/gridded/data.noaa.ersst.v5.html, and the ERSSTv5 uncertainties are available at
https://www.ncei.noaa.gov/pub/data/cmb/ersst/v5/ensemble.1854-2017/. No new data has been produced.
**Appendix A:  Critical slowing down indicators and significance calculation**

The restoring rate $\lambda$, the variance and the autocorrelation are calculated in the same manner as in B21. Each time series is first

non-linearly detrended using a running mean with a 50-year window. The edges are not removed, so the detrending method is

less certain at the first and last 25 years of the time series. The CSD indicators are then calculated in 60-year running windows.

The restoring rate of the underlying continuous dynamics is defined by the equation

$$dX_t = \lambda' X_t dt + \sigma dW_t,$$

which results from linearizing the dynamics around a given stable equilibrium, where $dW_t$ denotes the increments of a
Wiener process. Note that as in B21 the $\lambda$ plotted in this study is the numerical result of the regression of the discrete increments
$\Delta X_t = X_{t+1} - X_t$ against $X_t$, and so is related to the analytical, continuous $\lambda'$ by $\lambda = e^{\lambda'} - 1$ (when the timestep $\Delta t = 1$). As
the magnitude of $\lambda$ is immaterial in this study and we are only concerned with its increase or decrease, both definitions behave
similarly and are thus interchangeable for our purposes. In order to test the statistical significance of the linear slopes of CSD
indicators, Fourier surrogates are created from the detrended time series. These are calculated by taking the discrete Fourier
transform of the time series, multiplying by random phases and then taking the inverse Fourier transform. By the Wiener-





Khinchin theorem, the variance and autocorrelation function of wide-sense-stationary random processes are specified by the
squared amplitudes of the (discrete) Fourier transform. Thus the Fourier surrogates preserve the variance and autocorrelation
function of the original time series. For the calculations in the main text that use a 60 year window, 1000 Fourier surrogates are
generated for each time series. For the checks on the 50 and 70 year window lengths, 100 Fourier time surrogates are generated
for each time series for reasons of computational expense.
**Appendix B: Conservative probability calculation**
Since we use p≤0.05 as the condition for significance for each individual time series, for each time series there is a 0.05 chance
that it will be significant. The probability for significance in a number of time series then follows a binomial distribution, for
which the probability mass function is:
$$\mathrm{Pr}_b(k; n, p) = \binom{n}{k} p^k (1-p)^{n-k} \tag{B1}$$
This is the probability that the event with probability $p$ happens $k$ times out of the $n$ time series.

278       Out of the 1197 AMOC time series analysed in this study, 105 show a significant increase in a CSD indicator. If we regard

both the three CSD indicators and the three different AMOC strengths and fingerprints as independent, the probability of this
happening is $\mathrm{Pr}_b(105; 1197, 0.05) = 1.55e - 08$. However, neither the CSD indicators nor the AMOC strengths can be viewed
as independent (see Figs. C5, 6d, C6d and C7d). We thus have nine sets of 133 time series, each with their own number of
significant increases. For a set of 133 time series with $m$ significant increases, we can use the binomial distribution to calculate
the probability $p'$ of $m$ significances out of 133 as if the other sets don't exist: $p' = \mathrm{Pr}_b(m; 133, 0.05)$. In reality, the probability
is likely slightly higher than $p'$ because we have nine sets of time series, but these sets are dependent to an unknown extent, so
we cannot view the nine sets of time series as multiple experiments. The probability of the result for each strength-indicator or
index-indicator combination viewed alone are shown in Table B3.

287       We are also interested in the likelihood of the result for individual models. A model like CESM2 with 17 out of 90 significant

time series looks like it would be unlikely, but it is important to quantify this to some extent. Here again we consider the CSD
indicators and AMOC strengths and index separately, as there is no way to quantify their dependence. For a given CSD
indicator and AMOC strength we can then first calculate the probability for the model result considered alone. For model and
indicator-strength combination $i$ this is $p'_i = \mathrm{Pr}_b(l_i; n_i, 0.05)$, where $l_i$ is the number of significant increases out of $n_i$ time
series. However, as noted before, an event that would be unlikely if we only had one model could be likely to happen at least
once for 27 models. We thus need to consider how likely it is to have at least one event of probability $p'_i$ out of 27 instances. We
can do this here because for a given CSD indicator and AMOC strength we assume that the model results are independent. This
is again the binomial distribution from equation B1, but now with $n = 27$ and $p = p'_i$. We are now interested in the likelihood
of at least one occurrence of a $p'_i$ probability event, which is: $\mathrm{Pr}_b(k \geq 1; 27, p'_i) = 1 - \mathrm{Pr}_b(0; 27, p'_i)$. So combining this with the
equation for $p'_i$, we get:
$$\mathrm{Pr}_i(l_i) = 1 - \mathrm{Pr}_b(0; 27, \mathrm{Pr}_b(l_i; n_i, 0.05)) \tag{B2}$$





This is the probability that out of 27 models, we have at least one set with a probability $p_i'$ event (which for model $i$ is $l_i$ out
of $n_i$ significant time series). Of course since the value of $l_i$, $n_i$ and $p_i'$ will be different for the models, this is a conservative
calculation that applies only to each model individually - if we have multiple models with low $\mathrm{Pr}_i(l_i)$ the result in total is even
less likely. The model indicator-strength combinations for which $\mathrm{Pr}_i(l_i)$ is less than 0.05 are shown in Table B2.
These calculations assume that the CMIP6 models are completely independent from each other. This is not strictly true, and
in particular some dependence in the AMOC time series may arise from mulitple models using the same ocean component.
However, in our results there is no correlation between the ensemble members with lower p-values and the ocean component
(Fig. C13). In addition, the four models in Table B2 all use different ocean components. We can therefore assume that for our
purposes the CMIP6 models are independent.

|  | S26 | S35 | ASSTI | All three |
|---|---|---|---|---|
| **At least one** | 28 | 32 | 13 | 73 |
| **One** | 17 | 21 | 9 | 47 |
| **Two** | 10 | 6 | 4 | 20 |
| **Three** | 1 | 5 | 0 | 6 |

**Table B1.** Number of time series per streamfunction or SST Index with significant increases in the CSD indicators. Columns are the streamfunction strength latitude, AMOC SST Index and all three. Rows are the number of CSD indicators with a significant increase. So for example, the first cell from the top left counts the number of S26 timeseries that have at least one CSD indicator with a significant increase.

| Model | $\lambda$ S35 | AR1 S35 | AR1 S26 |
|---|---|---|---|
| **CanESM5** | 2.57e-02 | | |
| **CESM2** | 2.57e-02 | | 2.57e-02 |
| **HadGEM3-GC31-LL** | | 1.27e-02 | |
| **MIROC6** | | 2.57e-02 | |

**Table B2.** Probability of outcome amongst all models, for models where for a given CSD indicator and AMOC time series the outcome in CMIP6 is statistically unlikely to occur amongst 27 models (see methods).

|  | S26 | S35 | ASSTI |
|---|---|---|---|
| $\lambda$ | 4.17e-03 | 8.31e-05 | 1.04e-01 |
| **AR1** | 2.45e-04 | 6.76e-04 | 1.52e-01 |
| **Variance** | 9.19e-02 | 4.17e-03 | 1.59e-01 |

**Table B3.** Probability that a given set of time series has $m_i$ out of 133 significant increases, where $m_i$ is the number of significant increases in the indicator-index combination. This is calculated without considering the rest of the sets of time series.



**Appendix C: Extended figures and tables**

**Figure C1.** Same as Figure 1 but with the values for the ensemble members of each model plotted in a separate figure. Even though the aerosol forcing is the same within each model, there is still a large scatter of ensemble member slopes.





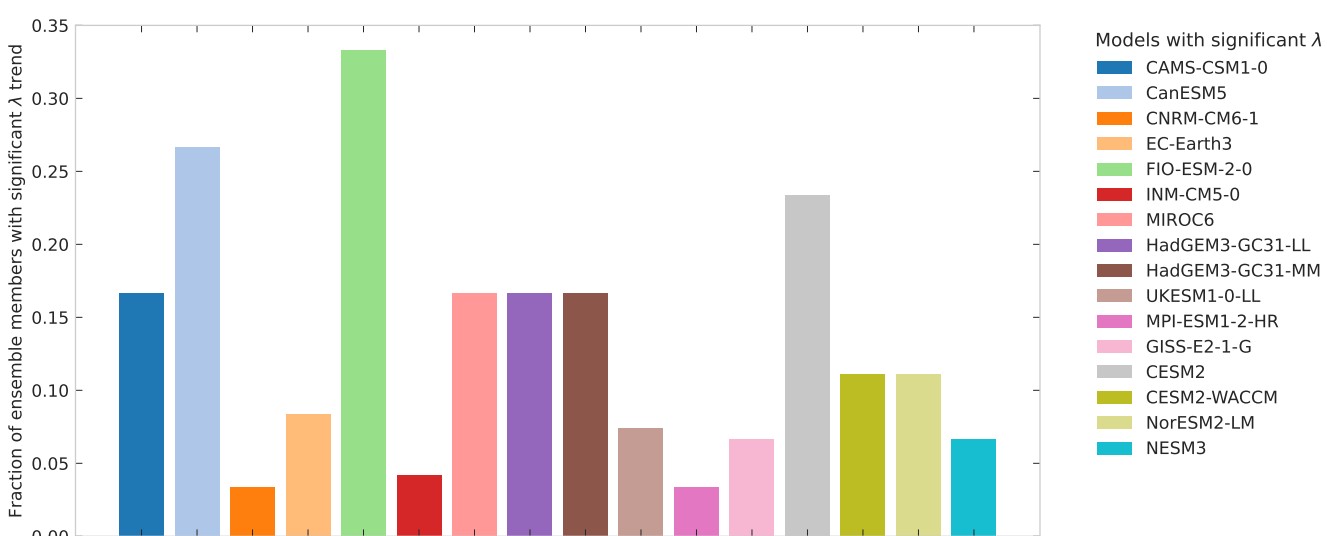

**Figure C2.** Fraction of all timeseries for a given model that have a significant increase in $\lambda$ (so both in streamfunction strength and SST Index). Only the models that do have such significant increases are shown, and the full list of these can be found in Table C4. Note that the colours for the models are different from other figures.





**Figure C3.** Same as Figure 4, but with the slices coloured according to the p-value of the corresponding $\lambda$. Darker blue is a smaller p-value and so more statistically significant.





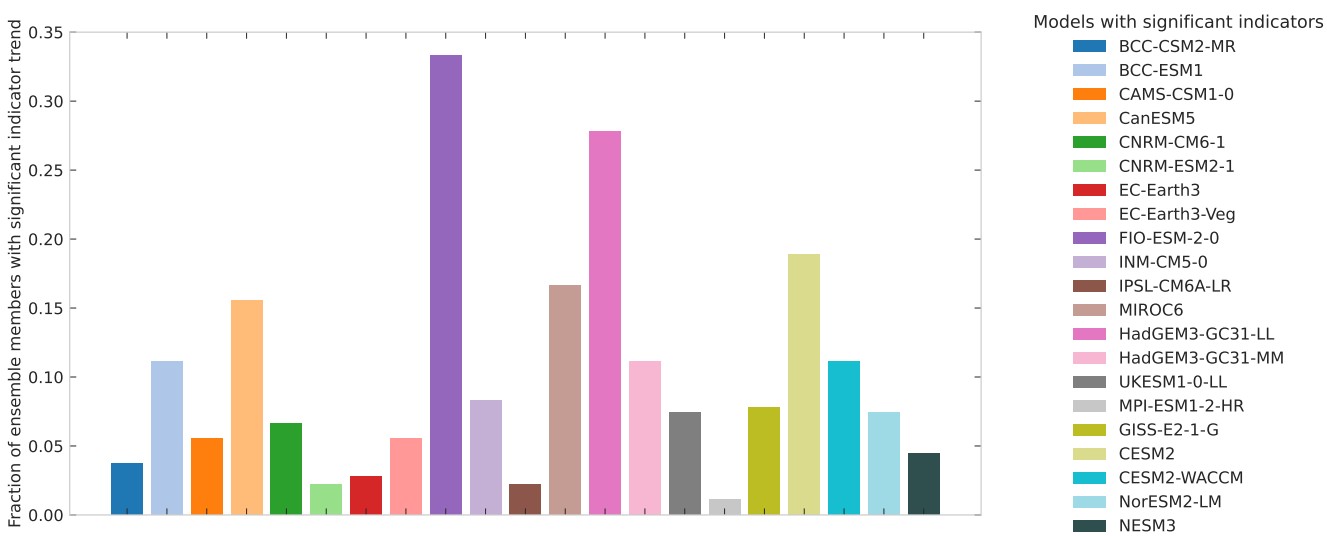

**Figure C4.** Fraction of timeseries for a given model that have a significant increase in any of the three CSD indicators. Only the models that do have such significant increases are shown, and the full list of these can be found in Tables C4, C5 and C6. Note that the colours for the models are different from other figures.





**Figure C5.** Same as Figure 6 but for the correlations between the p-values of different CSD indicators.







**Figure C6.** Same as Figure 6 for the AR1 instead of $\lambda$.





**Figure C7.** Same as Figure 6 for the variance instead of $\lambda$.





**Figure C8.** Same as Figure 4, but with a window size of 50 years.



**BCC-CSM2-MR | 3**   **BCC-ESM1 | 3**   **CAMS-CSM1-0 | 2**   **CanESM5 | 10**   **CNRM-CM6-1 | 10**   **CNRM-CM6-1-HR | 1**

**CNRM-ESM2-1 | 5**   **E3SM-1-1 | 1**   **EC-Earth3 | 4**   **EC-Earth3-Veg | 4**   **FIO-ESM-2-0 | 1**   **INM-CM4-8 | 1**

**INM-CM5-0 | 8**   **IPSL-CM6A-LR | 10**   **MIROC6 | 10**   **HadGEM3-GC31-LL | 4**   **HadGEM3-GC31-MM | 2**   **UKESM1-0-LL | 9**

**MPI-ESM1-2-HR | 10**   **MRI-ESM2-0 | 2**   **GISS-E2-1-G | 10**   **CESM2 | 10**   **CESM2-WACCM | 3**   **NorESM2-LM | 3**

**GFDL-CM4 | 1**   **NESM3 | 5**   **SAM0-UNICON | 1**

**Significantly increasing slope of $\lambda$ of:**
**AMOC Strength at 26.5°N**
**AMOC Strength at 35°N**
**AMOC SST Index**

**Figure C9.** Same as Figure 4, but with a window size of 70 years.



**Figure C10.** Same as Figure 5, but with a window size of 50 years.





**BCC-CSM2-MR | 3**    **BCC-ESM1 | 3**    **CAMS-CSM1-0 | 2**    **CanESM5 | 10**    **CNRM-CM6-1 | 10**    **CNRM-CM6-1-HR | 1**

**CNRM-ESM2-1 | 5**    **E3SM-1-1 | 1**    **EC-Earth3 | 4**    **EC-Earth3-Veg | 4**    **FIO-ESM-2-0 | 1**    **INM-CM4-8 | 1**

**INM-CM5-0 | 8**    **IPSL-CM6A-LR | 10**    **MIROC6 | 10**    **HadGEM3-GC31-LL | 4**    **HadGEM3-GC31-MM | 2**    **UKESM1-0-LL | 9**

**MPI-ESM1-2-HR | 10**    **MRI-ESM2-0 | 2**    **GISS-E2-1-G | 10**    **CESM2 | 10**    **CESM2-WACCM | 3**    **NorESM2-LM | 3**

**GFDL-CM4 | 1**    **NESM3 | 5**    **SAM0-UNICON | 1**

**Significantly increasing slope of:**

**All three indicators**
**Two out of three indicators**
**One out of three indicators**

**Figure C11.** Same as Figure 5, but with a window size of 70 years.





**Figure C12.** Same as Figure 6 for the ensemble mean values of each of the models.



**Figure C13.** The distribution of the ensemble member p-values for the $\lambda$ trend of the AMOC streamfunction strengths at 26.5°N and 35°N and the AMOC SST Index, plotted against the ocean component of the model. The models with lower p-values have a range of different ocean components.



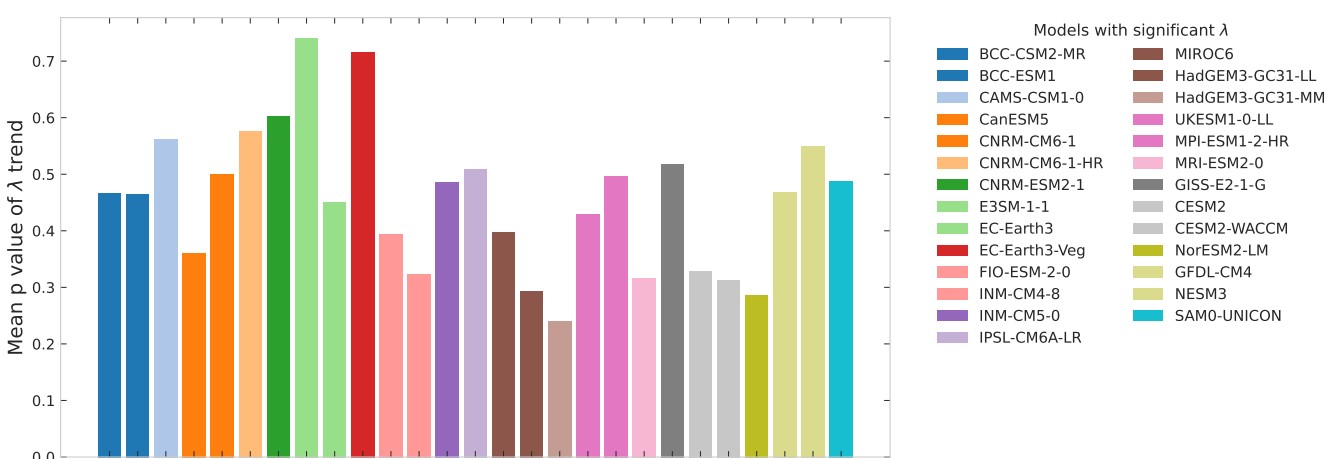

**Figure C14.** Mean p-value for all $\lambda$ timeseries of a given model. Note that the colours and shading of the models are different from other figures. Note that the models with highest indication of CSD are those with the *lowest* p-values.

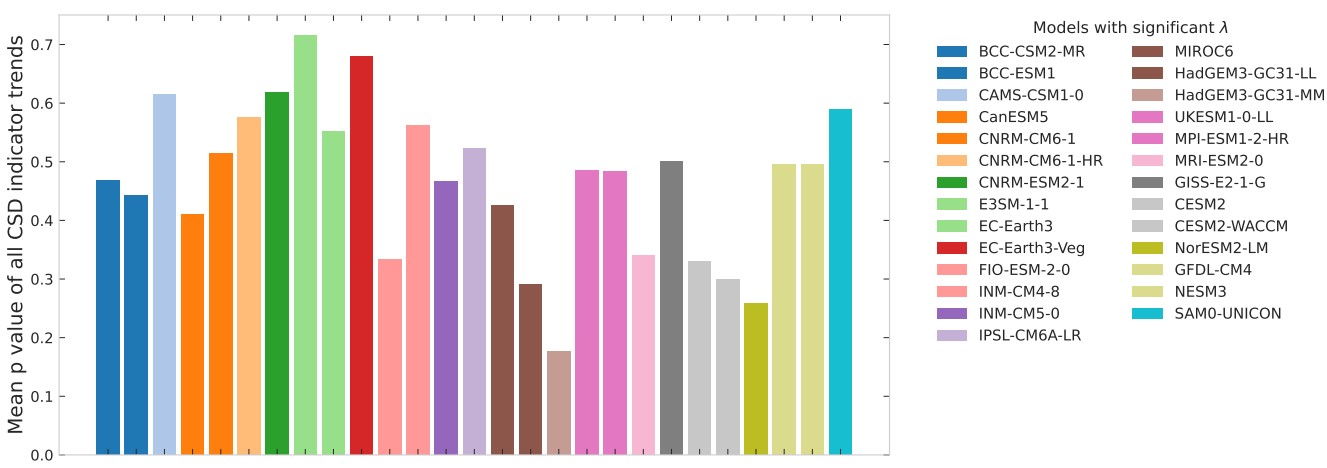

**Figure C15.** Mean p-value for all CSD indicator timeseries of a given model. Note that the colours and shading of the models are different from other figures.



| Model | Ensemble members |
|---|---|
| BCC-CSM2-MR | r1 r2 r3 |
| BCC-ESM1 | r1 r2 r3 |
| CAMS-CSM1-0 | r1 r2 |
| CanESM5 | r1 r2 r3 r4 r5 r6 r7 r8 r9 r10 |
| CNRM-CM6-1 | r1 r2 r3 r4 r5 r6 r7 r8 r9 r10 |
| CNRM-CM6-1-HR | r1 |
| CNRM-ESM2-1 | r1 r2 r3 r4 r5 |
| E3SM-1-1 | r1 |
| EC-Earth3 | r1 r2 r4 r7 |
| EC-Earth3-Veg | r1 r2 r3 r4 |
| FIO-ESM-2-0 | r1 |
| INM-CM4-8 | r1 |
| INM-CM5-0 | r1 r2 r5 r6 r7 r8 r9 r10 |
| IPSL-CM6A-LR | r1 r2 r3 r4 r5 r6 r7 r8 r9 r10 |
| MIROC6 | r1 r2 r3 r4 r5 r6 r7 r8 r9 r10 |
| HadGEM3-GC31-LL | r1 r2 r3 r4 |
| HadGEM3-GC31-MM | r1 r2 |
| UKESM1-0-LL | r2 r3 r4 r5 r6 r7 r8 r9 r10 |
| MPI-ESM1-2-HR | r1 r2 r3 r4 r5 r6 r7 r8 r9 r10 |
| MRI-ESM2-0 | r1 r5 |
| GISS-E2-1-G | r1 r2 r3 r4 r5 r6 r7 r8 r9 r10 |
| CESM2 | r1 r2 r3 r4 r5 r6 r7 r8 r9 r10 |
| CESM2-WACCM | r1 r2 r3 |
| NorESM2-LM | r1 r2 r3 |
| GFDL-CM4 | r1 |
| NESM3 | r1 r2 r3 r4 r5 |
| SAM0-UNICON | r1 |

**Table C1.** List of models and their ensemble members that are used in this study. All ensemble members for a listed rx are rxi1p1f1 except for CNRM-CM6-1, CNRM-ESM2-1, and UKESM1-0-LL which are rxi1p1f2 and HadGEM3-GC31-MM HadGEM3-GC31-LL which are rxi1p1f3.





|        | S26  | S35  | SSTI | Any  |
|--------|------|------|------|------|
| **Any**   | 24.0 | 35.0 | 21.0 | 80.0 |
| **One**   | 15.0 | 21.0 | 18.0 | 54.0 |
| **Two**   | 7.0  | 11.0 | 3.0  | 21.0 |
| **Three** | 2.0  | 3.0  | 0.0  | 5.0  |

**Table C2.** Same as Table B1 for a window size of 50 years.

|        | S26  | S35  | SSTI | Any  |
|--------|------|------|------|------|
| **Any**   | 31.0 | 40.0 | 14.0 | 85.0 |
| **One**   | 17.0 | 23.0 | 7.0  | 47.0 |
| **Two**   | 11.0 | 10.0 | 6.0  | 27.0 |
| **Three** | 3.0  | 7.0  | 1.0  | 11.0 |

**Table C3.** Same as Table B1 for a window size of 70 years.




| | r1 | r2 | r3 | r4 | r5 | r6 | r7 | r8 | r9 | r10 |
|---|---|---|---|---|---|---|---|---|---|---|
| **BCC-CSM2-MR** | | | | | | | | | | |
| **BCC-ESM1** | | | | | | | | | | |
| **CAMS-CSM1-0** | | S26.5 | | | | | | | | |
| **CanESM5** | | | S35 | | S35 | | S26.5 S35 | S26.5 | S35 | S26.5 ASSTI |
| **CNRM-CM6-1** | | | | | | | | S26.5 | | |
| **CNRM-CM6-1-HR** | | | | | | | | | | |
| **CNRM-ESM2-1** | | | | | | | | | | |
| **E3SM-1-1** | | | | | | | | | | |
| **EC-Earth3** | | S35 | | | | | | | | |
| **EC-Earth3-Veg** | | | | | | | | | | |
| **FIO-ESM-2-0** | S35 | | | | | | | | | |
| **INM-CM4-8** | | | | | | | | | | |
| **INM-CM5-0** | | S26.5 | | | | | | | | |
| **IPSL-CM6A-LR** | | | | | | | | | | |
| **MIROC6** | | | | | S35 | S35 | S26.5 | S26.5 S35 | | |
| **HadGEM3-GC31-LL** | | | S26.5 | S35 | | | | | | |
| **HadGEM3-GC31-MM** | S35 | | | | | | | | | |
| **UKESM1-0-LL** | | | | | S35 | | | S35 | | |
| **MPI-ESM1-2-HR** | | | | | | | | | | ASSTI |
| **MRI-ESM2-0** | | | | | | | | | | |
| **GISS-E2-1-G** | | ASSTI | ASSTI | | | | | | | |
| **CESM2** | | | S26.5 S35 | S35 | | | S26.5 | S26.5 S35 | S35 | |
| **CESM2-WACCM** | | | S35 | | | | | | | |
| **NorESM2-LM** | | | S26.5 | | | | | | | |
| **GFDL-CM4** | | | | | | | | | | |
| **NESM3** | | | | S26.5 | | | | | | |
| **SAM0-UNICON** | | | | | | | | | | |

**Table C4.** Significant increases of $\lambda$. The same abbreviations for the time series as Figure 6 are used. For each ensemble member the time series in which there is a significant increase is listed in the corresponding cell. Note that not all models have all 10 ensemble members - the list of members used can be seen in Table C1.




| | r1 | r2 | r3 | r4 | r5 | r6 | r7 | r8 | r9 | r10 |
|---|---|---|---|---|---|---|---|---|---|---|
| **BCC-CSM2-MR** | | | S35 | | | | | | | |
| **BCC-ESM1** | | | | | | | | | | |
| **CAMS-CSM1-0** | | | | | | | | | | |
| **CanESM5** | | | | ASSTI | | | S35 | S26.5 | S35 | S26.5 ASSTI |
| **CNRM-CM6-1** | | | | | | | | S26.5 ASSTI | | |
| **CNRM-CM6-1-HR** | | | | | | | | | | |
| **CNRM-ESM2-1** | | | | | | | | | | |
| **E3SM-1-1** | | | | | | | | | | |
| **EC-Earth3** | | | | | | | | | | |
| **EC-Earth3-Veg** | | | | | | | | | | |
| **FIO-ESM-2-0** | S35 | | | | | | | | | |
| **INM-CM4-8** | | | | | | | | | | |
| **INM-CM5-0** | S26.5 | | | | | | S26.5 | | | |
| **IPSL-CM6A-LR** | | | | | | | | | | |
| **MIROC6** | S35 ASSTI | S26.5 S35 | | | | | | S35 | | |
| **HadGEM3-GC31-LL** | | S35 ASSTI | S26.5 S35 | | | | | | | |
| **HadGEM3-GC31-MM** | S35 | | | S26.5 S35 ASSTI | | | | | | |
| **UKESM1-0-LL** | | S26.5 | | | | | | S35 | | S26.5 |
| **MPI-ESM1-2-HR** | | | | | | | | | | |
| **MRI-ESM2-0** | | | | | | | | | | |
| **GISS-E2-1-G** | | S26.5 | | ASSTI | | | | | | |
| **CESM2** | S26.5 | | S26.5 S35 | | | S26.5 | S35 | S26.5 S35 | | |
| **CESM2-WACCM** | | | | | | | | | | |
| **NorESM2-LM** | | | S26.5 | | | | | | | |
| **GFDL-CM4** | | | | | | | | | | |
| **NESM3** | | | | S26.5 | | | | | | |
| **SAM0-UNICON** | | | | | | | | | | |

**Table C5.** Significant increases in the autocorrelation. Same structure as Table C4.



| | r1 | r2 | r3 | r4 | r5 | r6 | r7 | r8 | r9 | r10 |
|---|---|---|---|---|---|---|---|---|---|---|
| **BCC-CSM2-MR** | | | | | | | | | | |
| **BCC-ESM1** | | | S26.5 S35 ASSTI | | | | | | | |
| **CAMS-CSM1-0** | | | | | | | | | | |
| **CanESM5** | | | | | | | | | | |
| **CNRM-CM6-1** | | | | | | | | S26.5 S35 ASSTI | | |
| **CNRM-CM6-1-HR** | | | | | | | | | | |
| **CNRM-ESM2-1** | | S26.5 | | | | | | | | |
| **E3SM-1-1** | | | | | | | | | | |
| **EC-Earth3** | | | | | | | | | | |
| **EC-Earth3-Veg** | | S26.5 S35 | | | | | | | | |
| **FIO-ESM-2-0** | S35 | | | | | | | | | |
| **INM-CM4-8** | | | | | | | | | | |
| **INM-CM5-0** | | S26.5 | | | | | S26.5 | | S35 | |
| **IPSL-CM6A-LR** | | | | S35 | | | ASSTI | | | |
| **MIROC6** | ASSTI | | S35 | | | | | S26.5 S35 | | |
| **HadGEM3-GC31-LL** | | | | S35 | | | | | | |
| **HadGEM3-GC31-MM** | | | | | | | | | | |
| **UKESM1-0-LL** | | | | | | | | S35 | | |
| **MPI-ESM1-2-HR** | | | | | | | | | | |
| **MRI-ESM2-0** | | | | | | | | | | |
| **GISS-E2-1-G** | | ASSTI | S35 | | ASSTI | | | | | |
| **CESM2** | | S26.5 | | | | | | | | |
| **CESM2-WACCM** | S26.5 S35 | | | | | | | S35 | S35 | |
| **NorESM2-LM** | | | | | | | | | | |
| **GFDL-CM4** | | | | | | | | | | |
| **NESM3** | | | | | | | | | | |
| **SAM0-UNICON** | | | | | | | | | | |

**Table C6.** Significant increases in the variance. Same structure as Table C4.



*Author contributions.* MBY, LB and NB conceived the study and designed it with contributions from SB. MBY and LB carried out the analysis. MBY, LB, SB and NB discussed results, and MBY wrote the paper with contributions from LB, SB and NB.

*Competing interests.* The authors declare that there are no competing interests.

*Acknowledgements.* MBY and NB acknowledge funding by the European Union's Horizon 2020 research and innovation programme under the Marie Sklodowska-Curie grant agreement No.956170. NB and SB acknowledge funding by the Volkswagen foundation. This is ClimTip contribution #X; the ClimTip project has received funding from the European Union's Horizon Europe research and innovation programme under grant agreement No. 101137601. LB acknowledges funding via the Tipping Points in the Earth System (TiPES) project that has received funding from the European Union's Horizon 2020 research and innovation programme under Grant Agreement No. 820970. In addition, the authors would like to thank Dr Matthew Menary for providing the CMIP6 AMOC data.



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
