# Peer review of "No critical slowing down in the Atlantic Overturning Circulation in historical CMIP6 simulations"

_EGUsphere, 2024_

## Author Comment (AC1)

**Reviewer #1**

The use of various 'Critical Slowing Down' (CSD) metrics as potential indicators of an approaching climate system tipping point or dynamical bifurcation has received a lot of attention recently. In particular the paper by Boers 2021 ('B21' in the present manuscript) showed an apparent increase (towards instability) in one of these CSD metrics as applied to observations of various proxies or 'fingerprints' of the AMOC – suggesting that the AMOC may have been getting closer to such a tipping point in recent decades. This paper examines the CMIP6 historical simulations, to see if the simulations bear any imprint of CSD as found in B21. It looks at three alternative CSD metrics, applied to three AMOC variables: the actual overturning strength at two latitudes, and a commonly used, SST-based AMOC fingerprint. Because model data is used the actual overturning timeseries are available as well as the indirect 'fingerprint' data, providing a check on the reliability of using the fingerprint as a proxy for the actual AMOC, as well as a check for consistent CSD indications across the three variables. The results are rather equivocal, with a few ensemble members of a few models showing indications of CSD in some, but not all, of the AMOC-related timeseries (as might be expected from random sampling). While it is hard to draw definite conclusions from the results, the authors do note that the SST fingerprint does not appear to show a lot of false positives for CSD, when compared with the actual (modelled) AMOC timeseries. This was not at all obvious *a priori*, and helps to build evidence that the SST-based fingerprint could be a useful tool in detection/early warning of AMOC tipping

This is a worthwhile study, the analysis has been carefully and thoroughly carried out and I found the presentation generally easy to follow. I liked the approach of looking at several variables as a way to test whether the CSD indicators are showing a consistent physical change. I have a number of fairly minor comments below, but overall I think the paper is worthy of final publication following some minor revisions in response to these.

Thank you for your review!

L 33-36, 65-68, 246-248. I think there is potentially some confusion here between a large weakening of the AMOC, which is what is described in these lines, and crossing a (fold?) bifurcation, which I think is what CSD is detecting. The AMOC could undergo a substantial weakening without crossing any bifurcation point (monostable throughout). Could CSD be used to detect the difference between these two situations? Lines 66-68 suggest that a few models do show a large AMOC weakening (albeit some time into the future). Are these the models that show CSD detection in their historical runs?

Thank you for highlighting this important point.

If we understand correctly, the reviewer is asking three questions: first, does CSD occur exclusively before a fold bifurcation, or could it occur in a case where the AMOC weakens but does not transition to an alternative stable state. Second, can CSD be used to distinguish between a linear weakening and a non-linear weakening. Third, do the models we know have a warming-induced AMOC collapse show CSD in the historical period. These questions are indeed not answered clearly enough in our work.

- The answer to the first question is no. In fact, as long as the linear stability weakens, CSD still occurs in systems even if they are not approaching a bifurcation point (see e.g. Kefi et al., 2012). We will rewrite parts of the introduction and conclusion to clarify this distinction.
- The answer to the second question is yes. Non-linearity is necessary for the occurrence of CSD, so a significant sign of CSD can be used to distinguish between a linear and a non-linear weakening.
- The answer to the third question is also no. Unfortunately, we cannot discuss this in detail in the manuscript as we only know about CMIP6 models which definitely have a warming-induced AMOC collapse in the ssp scenarios through personal correspondence. The reason we know about so few models is that, as we note in the paper, a weakening of the AMOC until 2100 is not necessarily a sign of a crossed or approaching bifurcation point. In the models that we know of, CESM2 and GISS-E2-1-G, the experiments are run far enough into the future that two distinct AMOC stable states emerge in the different scenarios, indicative of bistability. Referring to Figure 5, we can see that while CESM2 shows strong signs of CSD, GISS-E2-1-G does not. As we mention in the paper, this could just be due to the collapse happening after the 21st century.

Observations. Since this study takes advantage of the ability to use actual overturning from the models, could you also calculate the CSD indicators from the more 'oceanographically-based' AMOC estimate of Fraser and Cunningham https://agupubs.onlinelibrary.wiley.com/doi/full/10.1029/2021GL093893 ?

Thank you for this comment. There are two main reasons we chose not to include this fingerprint in our study:

- The first is that our study is not aimed at testing the validity of different fingerprints for AMOC CSD detection. If we had seen conclusive CSD in the AMOC streamfunction, we could then test different fingerprints for their validity. As it is, at most we could conclude that other fingerprints do not exhibit false positives. That result is less useful for fingerprints that have not been used to detect AMOC CSD, unlike the SPG SST-based fingerprint, which is primarily the one used in studies detecting CSD in the historical AMOC (Boers et al. 2021, Ben-Yami et al. 2023, Ditlevsen & Ditlevsen 2023).
- In addition, we believe that the estimate of Fraser and Cunningham may not be an ideal fingerprint for detecting CSD. As shown by Ben-Yami et al. 2023, the method used by the EN4 dataset to infill missing values introduces false trends in the statistical indicators, especially in the variance. Using

the full profiles of both salinity and temperature to calculate the transports will inevitably include the statistical biases of the preprocessing in the final product. As shown in Ben-Yami et al. 2023, these biases can be accounted for in the statistical significance calculation, however, this fingerprint needs to undergo a thorough uncertainty testing before it is used for an analysis of AMOC CSD.

We will add a new sentence to the text to explain why our study is not examining other AMOC fingerprints.

L 199-200. I'm not sure that this situation is so unlikely. We're looking at a time-varying dynamical system where there is natural internal variation on the same timescale as the forcing, so there will be some natural variability the effective equilibrium structure, which would cause variability in whether/when CSD was detected. Hence I'm not so sure that explanation (a) is less likely than (b). It might be possible to differentiate better between these possibilities through analysis of the multi-century pre-industrial control runs, but that would be another (substantial) study and I'm not suggesting that the authors need to do that in this paper. But in the absence of further evidence it feels to me as if both (a) and (b) are quite likely.

Thank you for pointing this out. It is indeed important to take into account the large internal variability of the AMOC, and we will remove the sentence about the unlikelihood of point (a).

L 235-245. I didn't really follow this paragraph (although I agree with the last sentence!). Surely (line 242) the evidence for the negative temperature feedback discussed comes from analysis of climate models? There are other negative feedbacks that seem to be stronger than this temperature feedback in some models, e.g. salinity advection by gyre processes, or atmospheric feedbacks (Jackson et al. https://link.springer.com/article/10.1007/s00382-016-3336-8), so, along with the fact that there are other processes driving the sub-polar SST (as noted in the manuscript), I'm not sure this paragraph is a strong argument for the SST index as an indicator of AMOC CSD/tipping.

Thank you for this important comment. Indeed, different models find different feedbacks for the AMOC, and different feedbacks connecting the AMOC and SPG. This was in fact our reasoning for the final sentence in the paragraph, but we agree that the rest of the discussion focused on one particular feedback too conclusively. We will rewrite the paragraph to say:

"Finally, it is not only uncertain how well the models represent the AMOC as a whole, but also how well they represent the connection between its components (Jackson et al., 2023b). For example, the extent, location and even existence of deep water formation in the SPG differs between models (Jackson et al., 2023b). These differences in the processes connecting the SPG SSTs to the AMOC could influence the AMOC SST Index's suitability to detect AMOC CSD, which may thus vary between models. Conclusive statements

on the validity of the AMOC SST Index would thus require an in-depth analysis of the physical processes, which is beyond the scope of this work."

**References**

Kéfi, S., Dakos, V., Scheffer, M., Van Nes, E.H. and Rietkerk, M. (2013), Early warning signals also precede non-catastrophic transitions. Oikos, 122: 641-648. https://doi.org/10.1111/j.1600-0706.2012.20838.x

N. Boers. Observation-based early-warning signals for a collapse of the Atlantic Meridional Overturning Circulation. Nature Climate Change, 11(8):680–688, 2021. ISSN 17586798. doi: 10.1038/s41558-021-01097-4. http://dx.doi.org/10.1038/s41558-021-01097-4.

Ben-Yami, M., Skiba, V., Bathiany, S., and Boers, N.: Uncertainties in critical slowing down indicators of observation-based fingerprints of the Atlantic Overturning Circulation, http://arxiv.org/abs/2303.06448, 2023.

P. D. Ditlevsen and S. Ditlevsen. Warning of a forthcoming collapse of the Atlantic meridional overturning circulation. (March):1–12, 2023. doi: 10.1038/s41467-023-39810-w. http://arxiv.org/abs/2304.09160.

---

## Author Comment (AC2)

**Reviewer #2**

I am unable to do a review of this paper and I am honestly very surprised to see a positive assessment from Anonymous Referee #1 because the paper lacks the Methods section. Maybe they had access to a different version of the paper than the one that is available online on this page?

The paper goes from the first section "Introduction" to the second section "Results" and the reader has no clue where the methods are. In fact, I was not able to find "Methods" in this paper and I have no idea which model simulations are used here. At some point at line 80 I am told to look at table C1 where the list of CMIP6 models would be found. Regrettably, here too I can't find what experiments are analyzed here, just a list of ensemble members. Perhaps historical since it's written in the title?

We apologize for the lack of a methods section – we will add a new one.

The methods are as follows:

- The simulations are indeed the historical CMIP6 simulations using the models and ensemble members from table C1.
- The time period is 1850-2014.
- The AMOC SST Index is calculated as the mean SSTs in the area between 41° and 60° N and 20° and 55° E, minus the mean global SSTs.
- The observational data is from the ERSSTv5 and HadCRUT5 datasets, and the uncertainty ranges are calculated from the uncertainty ensembles of these datasets following Ben-Yami et al. 2023.
- In the main text the CSD indicators are calculated in 60 year rolling windows.
- The restoring rate lambda is calculated by regressing the increments of the data against the data values using the GLSAR function from the statsmodel package in python, which is a Generalized Least Squares regression with AR covariance structure.

Ben-Yami, M., Skiba, V., Bathiany, S., and Boers, N.: Uncertainties in critical slowing down indicators of observation-based fingerprints of the Atlantic Overturning Circulation, http://arxiv.org/abs/2303.06448, 2023.

Here is a partial list of comments, but honestly I wasn't able to progress much into the paper since I have no idea what data are being analyzed here:

L24: remove "Roughly" from here

Done.

Fig. 1: how do you explain the outliers here? The INM models? What does the streamfunction look like for the two models?

As far as we can tell, there is nothing unusual about these model time series – there's simply a different trend for the stream function at the different latitudes (Fig. R1). This is definitely something worth investigating, but an in depth analysis of the INM models is beyond the scope of this paper.

[Figure]

Figure R1. AMOC timeseries of the three INM model ensemble members that are outliers in Figure 1 from the main text. The faint green (orange) lines are the timeseries of the AMOC at 26.5°N (35°N), and the solid straight lines are the fitted linear trends whose slopes are plotted in Figure 1 from the main text.

Fig. 2: not having a Methods section make this plot unreadable, what am I comparing models with? Which years are used for the observational data? Even without knowing what I am looking at, here there is nevertheless an important flaw: the fitted red slope is by eye influenced by outliers… you should present an analysis of linear fit that is not sensitive to outliers. Again not having methods for me it's impossible to figure out what the black line and gray bands mean for observations.

Thank you for pointing this out – this figure is indeed difficult to understand without a methods section. The years for the CMIP6 models are 1850-2014, and in Figure 2 are the same for ERSSTv5. The grey band is simply the minimum to maximum trend range out of the 1000 ensemble members from the ERSSTv5 uncertainty array. We hope the new methods section will clarify these questions.